Predicting surface abundance of federally threatened Jollyville Plateau Salamanders (Eurycea tonkawae) to inform management activities at a highly modified urban spring

Adcock Zachary C. 1 2
http://orcid.org/0000-0002-9793-7240 MacLaren Andrew R. 2 amaclaren89@gmail.com
Jones Ryan M. 2
Villamizar-Gomez Andrea 1
Wall Ashley E. 2
White IV Kemble 2
Forstner Michael R. J. 1
1 Department of Biology, Texas State University , San Marcos, Texas , United States
2 Cambrian Environmental , Austin, Texas , United States
Baxter-Gilbert James
Electronic publication date: 2022 May 2
Publication date: 2022
Volume: 10
Electronic Location ID: e13359
Received 2022 Jan 25; Accepted 2022 Apr 8
Copyright: © 2022 Adcock et al.
Copyright year: 2022
Copyright holder: Adcock et al.
License: This is an open access article distributed under the terms of the Creative Commons Attribution License, which permits unrestricted use, distribution, reproduction and adaptation in any medium and for any purpose provided that it is properly attributed. For attribution, the original author(s), title, publication source (PeerJ) and either DOI or URL of the article must be cited.
License URL: https://creativecommons.org/licenses/by/4.0/

Keywords: Conservation, Edwards aquifer, Habitat, Salamander, Spring, Threatened species, Urbanization, Abundance, Management

Funding: Williamson County Conservation Foundation This work was supported by the Williamson County Conservation Foundation. The funders had no role in study design, data collection and analysis, decision to publish, or preparation of the manuscript.

==============================
Urban expansion has contributed to the loss of habitat for range restricted species across the globe. Managing wildlife populations within these urban settings presents the challenge of balancing human and wildlife needs. Jollyville Plateau Salamanders (Eurycea tonkawae) are a range restricted, federally threatened, species of neotenic brook salamander endemic to central Texas. Almost the entire geographic range of E. tonkawae is embedded in the Austin, Cedar Park, and Round Rock metropolitan areas of Travis and Williamson counties, Texas. Among E. tonkawae occupied sites, Brushy Creek Spring has experienced some of the most extensive anthropogenic disturbance. Today the site consists of small groundwater outlets that emerge in the seams within a concrete culvert underlying a highway. Salamanders persist within this system though they are rarely detected. Here, we model the occurrence of salamanders within the surface habitat of Brushy Creek Spring using generalized linear models. In the absence of available data regarding the amount of water that is discharged from the spring, we use accumulated rainfall as a proxy for discharge to estimate salamander abundance. Additionally, we present evidence of reproduction, recruitment, and subterranean movement by E. tonkawae throughout this site. Infrastructure maintenance is inevitable at Brushy Creek Spring. We intend for our results to inform when maintenance should occur, i.e., during environmental conditions when salamanders are less likely to be observed in the surface habitat, to avoid unnecessary impacts to this federally threatened species.

Introduction

Urban expansion directly effects biodiversity in many ways, and one of the most concerning is the loss of habitat for range-restricted species (McDonald et al., 2018). Managing wildlife populations in urban settings presents the challenge of balancing human and wildlife needs (Aronson et al., 2017). This is especially difficult for threatened and endangered species management, as new development, or the maintenance of existing infrastructure in or near habitat, may result in “take” as defined by the U.S. Endangered Species Act of 1973 (as amended; United States, 1983). The U.S. Fish and Wildlife Service (USFWS) considers “take” as actions that “harass, harm, pursue, hunt, shoot, wound, kill, trap, capture, or collect, or attempt to engage in any such conduct” (United States, 1983; U.S. Fish & Wildlife Service, 2013a). Incidental take permits are commonly issued by the USFWS to allow take of a listed species from activities associated with an otherwise lawful project, but incidental take is rarely directly informed by population models (McGowan & Ryan, 2010).

Almost the entire range of Jollyville Plateau Salamanders (Eurycea tonkawae) is embedded in the Austin, Cedar Park, and Round Rock metropolitan areas of Travis and Williamson counties, Texas, USA (U.S. Fish & Wildlife Service, 2013a; Devitt et al., 2019). Conservation concern for the taxon began immediately upon its formal description due to its small geographic distribution in an urban environment (Price, Hillis & Chippindale, 1999; Chippindale et al., 2000). Eurycea tonkawae are neotenic, permanently aquatic, plethodontid salamanders restricted to groundwater-fed aquatic habitats, such as springs, spring-fed creeks, and caves, primarily in the northern segment of the Edwards Aquifer (Chippindale et al., 2000; Chippindale, 2005). Individuals are usually observed proximate to a spring outlet or a stream segment gaining groundwater (Sweet, 1982; Bowles, Sanders & Hansen, 2006), but can also be observed downstream of springs and in second order creeks (Bendik, McEntire & Sissel, 2016; Adcock et al., 2020). Typical surface habitat consists of shallow, flowing water with ample cover objects (e.g., rocks, leaf litter), substrate that provides interstitial spaces and access to subterranean water, and water chemistry associated with karst aquifers (Chippindale, 2005; Bowles, Sanders & Hansen, 2006; U.S. Fish & Wildlife Service, 2013a, 2013b).

The USFWS listed E. tonkawae as threatened in 2013 because of concerns regarding water quantity reduction, water quality degradation, and habitat loss due to urbanization (U.S. Fish & Wildlife Service, 2013a) and subsequently designated 32 critical habitat units (CHUs) for the taxon (U.S. Fish & Wildlife Service, 2013b). Eurycea tonkawae CHUs consist of both surface and subterranean components that are defined as 80 and 300 m radius circles, respectively, around the spring outlet (U.S. Fish & Wildlife Service, 2013b). The surface CHUs are restricted to aquatic areas up to the ordinary high-water lines (U.S. Fish & Wildlife Service, 2013b). An aerial review of the CHUs demonstrates that approximately 67% of the surface and 86% of the subsurface circles contain anthropogenic structures (e.g., buildings, roads), substantiating the urban nature of the taxon’s distribution. These structures will inevitably require maintenance for public safety, which may trigger federal consultations and evaluations of incidental take. Although the USFWS excluded anthropogenic structures from the surface CHUs, the subsurface CHUs extend below these structures, and construction or maintenance activities adjacent to or above the CHUs may have adverse indirect effects through runoff into the surface or subsurface aquatic environment (U.S. Fish & Wildlife Service, 2013b). These concerns are supported by previous work that determined E. tonkawae counts and density are negatively correlated with impervious cover, a metric of development and urbanization (Bowles, Sanders & Hansen, 2006; Bendik et al., 2014).

Brushy Creek Spring, aka Round Rock Spring, constitutes CHU 2 for E. tonkawae (U.S. Fish & Wildlife Service, 2013b). In a recent study of select Eurycea-occupied springs in the northern segment of the Edwards Aquifer, this site had the second highest amount of impervious cover in its watershed (Diaz et al., 2020). An office building, apartment buildings, and US 79 all occur directly adjacent to, above, and upstream of the spring (Chippindale et al., 2000; Chippindale, 2005). Additionally, the site was drastically altered in the early 2000’s when a large concrete culvert and gabion were installed over the original spring outlet to convey stormwater runoff from upgradient urban development. As such, Brushy Creek Spring is an excellent example of a location that will require perpetual infrastructure maintenance within an E. tonkawae CHU.

Eurycea salamanders were first documented from Brushy Creek Spring in 1948 and were considered Texas Salamanders (E. neotenes) until the formal description of E. tonkawae (Baker, 1961; Sweet, 1982; Chippindale et al., 2000). At this time, the only other known population of what are now considered E. tonkawae was from the nearby Kreinke Spring. It is possible that E. tonkawae were scarce within the surface habitat of Brushy Creek Spring prior to anthropogenic alterations. At the time of their discovery, only one voucher specimen was collected within Brushy Creek Spring, compared to 21 vouchered specimens at Kreinke Spring by the same researcher (Baker, 1961; Sweet, 1982; VertNet.org). Sweet (1978) conducted two surveys at this site between 1969 and 1974 without detecting salamanders, and only eight vouchers were collected between 1990 and 1994 (Chippindale et al., 2000; VertNet.org). The number of vouchered specimens may not necessarily reflect abundance; however, no alternative sources of information are available from Brushy Creek Spring prior to anthropogenic disturbance. At the time of federal listing, this site was considered the only known locality where E. tonkawae had been extirpated (SWCA Environmental Consultants, 2012, unpublished report), and prior to this study, the status of E. tonkawae at Brushy Creek Spring was unclear.

Other central Texas Eurycea salamanders are known to occur or increase in density and abundance in the surface habitat after periods of rainfall and/or increases in groundwater discharge (Gillespie, 2011; Tovar & Solis, 2013, Bendik & Dries, 2018). Urbanization may result in the development of a shallow pseudo-karst (e.g., tunnels, conduits, utility networks), which can evolve rapidly and dominate water flow and transport (Garcia-Fresca, 2007). Slade, Dorsey & Stewart (1986) demonstrates that within the Austin area, surface recharge (e.g., rainfall) influences discharge, sometimes rapidly, with discharge increasing only days after rain events. In Austin-area watersheds, rainfall and the associated runoff decreases the specific conductance (SC) of water within spring runs. When runoff is absent and groundwater dominates flow, SC typically rises (Johns, 2006). However, increased groundwater discharge may decrease SC when it reduces aquifer residence time (Gillespie, 2011). Dissolved oxygen (DO) typically decreases as groundwater discharge decreases (Gillespie, 2011). Rainfall, SC, and DO may serve as proxies for groundwater discharge volume when this metric is unknown. To the best of our knowledge, no study of the dynamic relationship between aquifer recharge from rainfall and groundwater discharge has been conducted at Brushy Creek Spring.

Here, we report the findings of 7 years of surveys at Brushy Creek Spring. We first sought to determine if E. tonkawae still occur at this site, and if so, where they occur within the CHU. Then, enabled by the detection of salamanders, we estimated capture probability and survival rate. Next, we tested for differences in water chemistry (i.e., temperature, pH, DO, and SC) across habitat components. Finally, we modeled environmental covariates (i.e., season, rainfall) that predict E. tonkawae relative abundance in the surface habitat in order to identify periods when maintenance activities are less likely to disturb salamanders. We hypothesized that pulses in rainfall are correlated to salamander abundance at Brushy Creek Spring, while considering that these pulses may be delayed in their influence due to the unknown size of the underground karst (or pseudo-karst) system which provides this site with groundwater.

Materials and Methods

Study site

Brushy Creek Spring is located 1.94 km northeast of downtown Round Rock, Texas (Fig. 1). The site is characterized by two major components, a large, concrete culvert system and a spring run, which we define here as a body of flowing water that is primarily fed by a spring or group of springs. The culvert system consists of three tunnels that run under US 79 and empty into a two-sided box culvert. The culvert is owned by the City of Round Rock but occurs entirely within a Texas Department of Transportation right of way, who have provided us access and approval to conduct research at this site. During construction, polyvinyl chloride (PVC) pipes were installed to divert groundwater from the original spring location into the box culvert (C. Newnam, 2015, personal communication). Currently, groundwater emerges from the PVC pipes, seams in the concrete culvert tunnel junctions under US 79, and at seams and cracks at the concrete culvert tunnel and box culvert interface. The number of seams and cracks discharging groundwater changes with fluctuating aquifer levels. Spring water flows in a shallow sheet, often less than 1 cm deep, for 20 m inside the box culvert, over and through a gabion structure, and into a deep pool before constricting into a spring run that travels approximately 30 m before waterfalling into Brushy Creek (Fig. 2). The spring run exemplifies typical E. tonkawae habitat, containing shallow, flowing water with abundant rocks and gravel substrate (Bowles, Sanders & Hansen, 2006; U.S. Fish & Wildlife Service, 2013a, 2013b). In contrast, potential cover objects within the culvert system change in response to flash floods. The system receives considerable stormwater from the upgradient development, and these floods both deposit and flush potential salamander cover objects from the culvert. Cover objects can include rocks but also litter (e.g., shopping carts, bottles, cans, clothing) and landscaping debris (e.g., branches, grass clippings). Litter and debris frequently get caught between the baffle walls in the box culvert (see Fig. 2C) forming temporary dams with small pools.

Figure 1 Location of Brushy Creek Spring.

Brushy Creek Spring (red star) in the context of the urban matrix of Round Rock, Williamson County, Texas, USA.

Figure 2 Descriptive site images of Brushy Creek Spring.

(A) Aerial image (Google Earth imagery) of Brushy Creek Spring in Round Rock, Williamson County, Texas. Stars designate the locations of photographs B–E. All photographs face upstream. (B) Junction of culvert tunnel under US79 and box culvert with groundwater discharging from the PVC pipe, seams in the culvert tunnel junctions, and at seams and cracks at the culvert tunnel and box culvert interface. (C) Right side of the box culvert upstream of the gabion with shallow, spring water sheet flowing across concrete bottom. (D) Downstream entrance of box culvert, gabion, and pond. (E) Spring run downstream of gabion (Photographed by Zach Adcock).

Salamander surveys

We conducted monthly to bi-monthly E. tonkawae survey events at Brushy Creek Spring from May 2014 to July 2021. On a few occasions, we were unable to survey during a scheduled month because of high stormwater flow. We searched for E. tonkawae under and in available cover objects and recorded the number of objects searched and the time spent surveying. We surveyed just the spring run portion of the site from May 2014 to March 2015, as this appeared to be the most appropriate habitat for E. tonkawae. From April 2015 to January 2018, we surveyed both the spring run and box culvert during each survey event. Right of entry to the spring run was unavailable after January 2018, but we maintained culvert surveys throughout the remainder of the study. We surveyed the spring run 28 times and the culvert 39 times over the course of this study.

We attempted capture of each observed E. tonkawae using aquarium nets, and we recorded body and head photographs on a standardized grid background with the salamander in a water-filled dish. Pigmentation patterns on the head were used to identify recaptured salamanders using Wild-ID photographic recognition software (Bolger et al., 2012; Bendik et al., 2013). We determined gravidity of all captured E. tonkawae by visually checking for oocytes through the salamander’s translucent venter (Fig. 3; Gillette & Peterson, 2001; Pierce, McEntire & Wall, 2014). We measured snout-vent length (SVL) and total length (TL) of all captures. Salamanders were either measured by hand using a metric dial caliper, or digitally using the software ImageJ, where size standardization is taken from the gridded background of all salamander photos. We conducted this study in compliance with Federal Fish and Wildlife Permits TE039544-1 and TE37416B-0, Texas Parks and Wildlife Scientific Permits SPR-0102-191 and SPR-0319-056, and Texas State University Institutional Animal Care and Use Committee (IACUC) permits 1202_0123_02 and 0417_0513_07.

Figure 3 Gravid female Jollyville Plateau Salamander.

(A) Jollyville Plateau Salamander (Eurycea tonkawae) from Brushy Creek Spring. (B) Oocytes are visible through the salamander’s translucent venter (Photographed by Ryan Jones).

We measured water temperature (°C), pH, DO (mg/L), and SC (µS/cm) during each survey. These parameters were collected using the following instruments, based on their availability to surveyors: HI 9828 multiparameter probe (Hannah Instruments, Smithfield, RI, USA), Manta2® multiprobe (Eureka Water Probes, Austin, TX, USA), Com-100 (HM Digital, Culver City, CA, USA), EcoTestr pH2 (Oakton Instruments, Vernon Hills, IL, USA), or HI 9147 (Hannah Instruments, Woonsocket, RI, USA). We downloaded daily precipitation accumulation from the National Oceanic and Atmospheric Administration (NOAA) stations US1TXWM0039, US1TXWM0178, US1TXWM0195, and US1TXWM0219 in Round Rock, Texas. We averaged the daily values among stations when data overlapped.

Capture-mark-recapture analyses

We constructed all capture-mark-recapture models in program R (R Version 3.6.1; R Development Core Team, 2018) using the package ‘Rmark’ (Laake & Rexstad, 2008; Laake, 2013) which calls program MARK (White & Burnham, 1999). We used a Cormack-Jolly-Seber (CJS) model (Cormack, 1964; Jolly, 1965; Seber, 1965) to estimate capture probability and survival rate of all captures (n = 27). Under this formulation, capture probability (pi) is defined as the probability that a marked animal in the study population at sampling period i (n = 39) is captured or observed during period i (Williams, Nichols & Conroy, 2002). Survival rate, also referred to as apparent survival, ( φi) is defined as the probability that a marked animal in the study population at sampling period i survives until period i + 1 and does not permanently emigrate (Williams, Nichols & Conroy, 2002). Because of a limited sample size, and rare occurrence of recaptures, we estimated a single CJS model with constant detection probability and survival, the null model. Throughout our study two individuals were observed but evaded capture. It is important to note that these individuals are included in our models predicting abundance in response to rainfall accumulation but cannot be included in our CJS model. We confirmed goodness-of-fit using package ‘R2ucare’ (Gimenez et al., 2018).

Environmental variable selection

We sought to test the influence of rainfall accumulation, as a proxy for groundwater discharge, on salamander observations. To do so, we calculated 30, 60, and 90 days of accumulated rainfall prior to each survey event. The amount of time that occurs between a rainfall event and increased groundwater discharge depends upon the amount of rainfall, current aquifer levels, the size of the local karst system, and retention time (groundwater age) in the karst system. This has been measured in other portions of the Edwards Aquifer of Texas and can range from less than 1 year up to several decades (Hunt et al., 2012). Properties of the local karst system are not known at Brushy Creek Spring. To account for this unknown, we calculated 30, 60, and 90-day accumulations 1, 2, and 3 months previous to each survey event. For example, 90 days of accumulation, occurring three months previous to surveys provides the total rainfall accumulated 91–180 days prior to each survey event. In total, 12 ranges of rainfall accumulation were tested.

Central Texas Eurycea salamander surface abundance is known to increase throughout the spring, peak in the summer, and decrease in autumn and winter (Bowles, Sanders & Hansen, 2006; Pierce et al., 2010; Bendik, 2017). To account for this phenology, we included the quadratic effect of day-of-year (day2) as a predictor within each model (Kéry & Royle, 2016; Edwards & Crone, 2021). We included the lower-order term of “day” to adhere to the rules of marginality (McCullagh & Nelder, 1989; Kéry & Royle, 2016).

Additionally, we sought to investigate if changes in water conditions (i.e., temperature, pH, DO, and SC) caused by increasing groundwater discharge, rather than base flow or overland runoff, influenced salamander relative abundance. However, we were concerned with overfitting our models due to our limited sample size. Prior to utilizing water condition parameters as additional predictors within our generalized linear models, we tested for differences among these metrics with respect to salamander occurrence (present vs absent) within the culvert, using Welch’s two sample t-tests conducted in Program R (package ‘stats’). We also tested for differences in overall water conditions between the culvert and the spring run portions of the site (Table 1). We removed two surveys with outliers: one DO spike caused by a burst municipal water main which contributed overland runoff during a survey event and one event where equipment failure yielded untrustworthy estimates of SC (see Supplemental R Code).

Table 1 Water conditions are Brushy Creek Spring.

Location	Culvert	Spring run	
Salamander occurrence	Present (n = 16)	Absent (n = 18)	Present (n = 1)	Absent (n = 17)	
Temperature (°C)	21.92 ± 1.89	22.15 ± 2.94	20.56 ± 0.0	21.74 ± 2.66	
pH	7.04 ± 0.44	7.21 ± 0.35	7.84 ± 0.0	7.86 ± 0.30	
Dissolved oxygen (mg/L)	4.99 ± 1.29	6.52 ± 3.76	8.47 ± 0.0	6.57 ± 1.37	
Conductivity (µS/cm)	917.08 ± 47.43	877.37 ± 90.67	916.4 ± 0.0	819.98 ± 71.36	
Note:

Water conditions (µ ± σ) at the spring outlets inside of the culvert system (culvert) or in the spring run downstream of the spring outlets and culvert (run). Sample sizes calculated after missing values and outliers (n = 2) were removed.

Water temperature is subject to the influence of ambient air temperature, which peaks mid-year, mirroring the influence of day-of-year in our analysis. Although water temperature and salamander abundance are both reported to peak simultaneously, reproductive phenology is thought to be the cause for increase in surface salamander counts (Bendik, 2017), rather than coincidental increases in water temperatures. Further, water conditions within the culvert showed little variation within the duration of our study period (Tables 1 and 2, Fig. 4). Invariance can cause issues with model fit, especially in examples using small sample sizes (Warton et al., 2016). For these reasons, and on the basis of the results of our t-tests, we chose to remove all water chemistry parameters as predictors in our GLMs.

Table 2 Results for tests of water condition differences by location and occurrence.

Test description	Present (n = 16) vs Absent (n = 18)	Culvert (n = 34) vs Spring run (n = 18)	
Statistic	DF	t	P	DF	t	P	
Temperature (°C)	29.31	−0.279	0.783	33.33	−0.491	0.627	
pH	28.71	−1.281	0.210	44.85	7.625	<0.001	
Dissolved oxygen (mg/L)	27.14	−1.277	0.212	45.99	1.526	0.134	
Conductivity (µS/cm)	29.52	1.329	0.194	35.75	−3.292	0.002	
Note:

Degrees of freedom (DF), test statistic value (t), and P-value (P) for each water condition parameter measured. Data used to compare between salamander presence and absence was restricted to those collected from the culvert only. All data were used to test for differences between the culvert and spring run at Brushy Creek Spring. Bold text indicates statistical significance at P ≤ 0.05.

Figure 4 Water chemistry by occurrence and location.

Mean and 95% confidence interval for water chemistry metrics (measured at Brushy Creek Spring with respect to salamander occurrence (presence or absence) and location (culvert or spring run)).

We additionally excluded data collected from the spring run because only a single salamander was captured within this portion of the site during our study. Different variables may restrict salamander occurrence in this portion of the site (see Discussion) compared to those that influence salamander relative abundance in the culvert, thus making the spring run data not suitable to test our general hypothesis.

Environmental model analyses

We sought to model the environmental parameters that predict E. tonkawae relative abundance in the surface habitat. Count data such as these are generally analyzed using Poisson regression (O’Hara & Kotze, 2010). We performed a chi-squared goodness of fit test (package ‘vcd’) to test whether our data conform to a Poisson distribution ( χ2 = 2.61, P = 0.27). We then fit a suite of Poisson regressions via generalized linear models in Program R (GLMs; package ‘stats’). Ecological processes can often result in overdispersion, where variation is greater than is expected given a pure Poisson process (Lindén & Mäntyniemi, 2011). Because our initial goodness-of-fit test demonstrated a lack of fit, we tested for overdispersion (package ‘msme’). For all models the dispersion parameter (i.e., Pearson- χ2/Residual degrees of freedom) was greater than 1 and less than 2, indicating overdispersion (Kéry & Royle, 2016). Thus, we proceeded by refitting models using the negative binomial error distribution (Ver Hoef & Boveng, 2007). We fit all models using E. tonkawae observations (i.e., counts) as the response variable, including individuals observed but not captured.

After reducing the number of predictors, we fit a model for each rainfall accumulation range, with the additional predictors of day-of-year, and day-of-year2, resulting in 12 total models. Each model included a log offset of effort to control for count variations that may be due to differential survey effort among location or events (Kéry & Royle, 2016). We considered the model with the lowest Akaike’s Information Criterion value adjusted for small sample sizes (AICc) and the greatest model weight (ωi) as the best supported model, and we determined the importance of covariates with a Wald Z-test (Burnham & Anderson, 2002; Bolker et al., 2009).

We predicted salamander counts over a range of 0–60 cm of accumulated rainfall for the best fit model. For predictions, we held the log offset of effort constant at the mean number of objects searched (n = 172.3) and day-of-year constant at the mean value (194.74).

Results

We surveyed the spring run 28 times, for a total 725 person-minutes and 6,066 searched cover objects. We surveyed the culvert 39 times, for a total 1,438 person-minutes and 6,720 searched cover objects. We detected 24 individual E. tonkawae a total of 27 times (three recaptures) from 2014 to 2021. Captures ranged from 0 to 3 individuals per survey. Two individual salamanders avoided capture by retreating into groundwater outlets in the cracks in the culvert. We treat these individuals as unique, that is, not a recapture, because their identity cannot be verified. Two recaptures occurred in the survey event immediately following initial capture (i.e., the following month), one individual was recaptured a full year after first being detected, and no animals were recaptured more than once. Salamander SVL ranged from 12.0–41.3 mm with a mean of 29.5 ± 7.7 mm, and TL ranged from 20.5–84.5 mm with a mean of 56.6 ± 16.7 mm. We captured three gravid females (Fig. 3), and two juvenile-sized individuals (<15.0 mm SVL). The three gravid females were observed with 8, 17, and 24 oocytes visible through their venter.

We observed a single individual E. tonkawae on one occasion (no recaptures) downstream of the culvert system and gabion. This salamander was approximately 37.5 m downstream of the nearest spring outlet in rocks and gravel on the downstream edge of the pond before it constricts into the spring run. All other observations occurred on the concrete culvert apron (n = 1), in gaps between culvert tunnel junctions (n = 5), or inside of the culvert system under rocks, litter, or debris on the concrete floor (n = 21). Most E. tonkawae (n = 20) were observed within 1.5 m of a spring discharge outlet at the PVC diversion pipes or cracks and seams in the concrete (Fig. 5).

Figure 5 Jollyville Plateau Salamander (Eurycea tonkawae) capture location at Brushy Creek Spring.

(A) Groundwater discharges from cracks in the concrete along the culvert wall and floor junction and from the nearby spring diversion PVC pipe. White square outlines the area in photo (B). One E. tonkawae partially covered by a large cobble (Photographed by Zach Adcock).

Capture-mark-recapture model

Our CJS model estimated capture probability (pi) to be 0.063 ± 0.054 (lcl = 0.011, ucl = 0.289) and survival rate ( φi) = 0.68 ± 0.159 (lcl = 0.337, ucl = 0.899). Goodness-of-fit tests demonstrated no lack-of-fit (P = 1). These estimates are highly influenced by our discovery of one individual which survived a full year before being recaptured. With this individual excluded, estimates converge upon their boundaries of pi = 1 and φi = 0 (see Supplemental R Code).

Environmental covariate models

The AICc model selection resulted in two competing models, that is, the difference between their AICc values is less than 2 (Table 3). In each of these models, rainfall accumulation was the only significant predictor of salamander abundance (Table 4). The two most favored models both reflect a 90-day accumulation of rainfall; 31–120, and 1–90 days of cumulative rainfall, respectively. The top model reflects rainfall delayed by a single month (31–120 days), compared to the closest competing model which reflects recent rainfall (1–90 days). Under both models, salamander counts increase as rainfall accumulation increases (Table 4). Although seasonal shifts in surface abundance are well documented (Bowles, Sanders & Hansen, 2006; Pierce et al., 2010; Bendik, 2017), the quadratic effect of day-of-year was not significant in our top two models (Table 4). However, these predictors appear near significance, and it may be that our limited sample size was simply not sufficient to realize this well-known seasonal phenology.

Table 3 Results of model selection.

Model	Rainfall accumulation	K	AICc	ΔAICc	ω i	Pseudo R2	
6	31–120	5	93.046	0.000	0.327	0.252	
3	1–90	5	94.695	1.649	0.143	0.215	
2	1–60	5	95.179	2.133	0.112	0.204	
5	31–90	5	95.283	2.237	0.107	0.201	
4	31–60	5	95.612	2.566	0.091	0.193	
8	61–120	5	96.379	3.333	0.062	0.175	
7	61–90	5	96.722	3.675	0.052	0.167	
9	61–150	5	98.212	5.166	0.025	0.131	
10	91–120	5	98.418	5.372	0.022	0.125	
12	91–180	5	98.457	5.411	0.022	0.124	
1	1–30	5	98.656	5.61	0.02	0.119	
11	91–150	5	98.885	5.839	0.018	0.114	
Note:

Top generalized linear models (GLMs) assessing the response variable of Jollyville Plateau Salamander (Eurycea tonkawae) counts at Brushy Creek Spring as predicted by the accumulation of rainfall over the specified days. The model, number of parameters (K), Akaike’s Information Criterion value adjusted for small sample sizes (AICc), difference in AICc from the top model (∆AICc), model weight (ωi), and the Cragg-Uhler-Nagelkerke pseudo R2 are presented.

Table 4 Summary and main effects of top competing models.

Parameter	Estimate	Standard error	Z -value	P	
Model 6					
Intercept	−7.036	0.735	−9.579	<0.001	
Day	0.011	0.008	1.532	0.126	
Day2	−3.995E−5	2.136E−5	−1.871	0.061	
Rain accumulation 31–120	0.038	0.015	2.551	0.011	
Model 3					
Intercept	−6.777	0.716	−2.2620	<0.001	
Day	0.012	0.008	1.4170	0.160	
Day2	−3.880E−5	2.171E−5	−1.6840	0.074	
Rain accumulation 1–90	0.033	0.016	2.0160	0.032	
Note:

Model summaries of all competing (∆AICc < 2) generalized linear models assessing the response variable of Jollyville Plateau Salamander (Eurycea tonkawae) counts to environmental predictors at Brushy Creek Spring. Bold text indicates statistical significance at P ≤ 0.05.

The mean predictions made using our top model estimate that approximately 29 cm of accumulated rainfall 31–120 days prior to survey are needed to observe one E. tonkawae inside of the culvert at Brushy Creek Spring (Fig. 6). The competing model’s mean prediction was approximately 25 cm of accumulated rainfall 1–90 days prior to survey. It is worth noting that these predictions are probabilistic, and that it is possible to observe one E. tonkawae with less rainfall accumulation, as indicated by the confidence intervals illustrated in Fig. 6.

Figure 6 Number of salamander detections predicted by rainfall accumulation.

(Top) Jollyville Plateau Salamander (Eurycea tonkawae) detections (circles) and predicted curve (black line) over a range of 0–60 cm of accumulated rainfall at Brushy Creek Spring for the best fit generalized linear model (Tables 3 and 4). The red line demarcates one salamander detection. The log offset of effort was held constant at the mean number of objects searched (172.3) and day-of-year as held constant at the mean day-of-year surveyed (194.74) within the culvert throughout the 39 surveys conducted for this study. (Bottom) Red bars represent the number of Jollyville Plateau Salamanders observed by date (secondary y-axis), and the black line represents rain accumulation 31–120 days prior to each date (primary y-axis).

Discussion

This study provides the first regular detections of E. tonkawae in the surface habitat at Brushy Creek Spring. Additionally, we document the first salamander observation in 21 years (1994–2015; Chippindale et al., 2000; VertNet.org) and the first since it was suggested that local extirpation had likely occurred (SWCA Environmental Consultants, 2012, unpublished report). We conducted surveys across six reproductive seasons (Bendik, 2017), observing three gravid females and two juvenile individuals, indicating successful reproduction at this site. These gravid females were observed during winter months (November, December, and January) and juvenile individuals during summer months (June and July) in general accordance with the reported reproductive phenology for this species (Bendik, 2017).

Our analysis of environmental predictors of salamander counts demonstrates that the accumulation of rainfall 31–120 days prior to survey best predicts E. tonkawae relative abundance. This is congruent with reports that abundance is correlated to lagged rainfall for other central Texas Eurycea taxa (Gillespie, 2011; Krejca et al., 2017). We used rainfall as a substitute for spring discharge because measuring discharge is not practical at Brushy Creek Spring due to multiple small discharge points scattered throughout the wide culvert. We acknowledge that an interaction between rainfall and aquifer water level would likely improve the fit of our models and reduce the estimated confidence intervals, but aquifer data are not available on a monthly scale for the duration of this project.

Ecological studies, and more specifically herpetological studies, often fail to account for imperfect detection (Ficetola, 2015; Kellner & Swihart, 2014). For monitoring efforts that are not designed around this explicit goal, accounting for imperfect detection can be challenging (Kéry & Schmidt, 2008). We attempted to model the probability of detecting marked individuals, but we lacked sufficient recaptures to do so. We do not present our capture-mark-recapture results as defensible estimates, given that our sample size limited us to fitting an overly simplistic model. We recaptured only three individuals throughout the duration of our seven-year study. Fitting capture-mark-recapture models to limited detections leads to uncertain parameter estimates (Durso, Willson & Winne, 2011; Mazerolle et al., 2007). The inclusion of a single recapture after a one-year period reversed our detection and survival estimates, indicating they are likely spurious. We hypothesize that survival may be low for salamanders in the surface habitat unless they are able to locate conduits that provide access back into the subsurface environment, as we observed in our study. It is important to note that our estimate of apparent survival cannot be disentangled from death (e.g., trapped in the surface habitat) or permanent emigration (e.g., retreating to the subterrain), and that these events would equally reduce estimated apparent survival. Additionally, we would expect detection to be high in the surface habitat of the culvert because there are few cover objects and the concrete floor limits subsurface escape. We note at other Eurycea occupied sites with similar concrete streambeds capture probability has been estimated as high as 0.82 (Bendik et al., 2021).

For the salamander with the 1-year recapture timeframe, we initially captured the individual in a pool formed behind a debris pile in the box culvert within 1 m of a spring outlet. We recaptured the individual in a seam between the concrete culvert tunnels approximately 20 m upstream of the original capture location. The floor of the culvert tunnel and box culvert are disjunct (Fig. 2B), and this individual almost certainly had to travel through the subterranean environment to this upstream location. This implies that, in spite of anthropogenic modifications, salamanders are able to seek refuge and survive in the subterranean karst or pseudo-karst at Brushy Creek Spring as in natural systems (Bendik & Gluesenkamp, 2013). Further, a recent study also found that Brushy Creek Spring has an exceptional aquatic invertebrate community score (Diaz et al., 2020), which is a measure of aquatic life use, and indicates a stable food source for salamanders at this site.

During our study we only observed a single individual within the spring run downstream of the box culvert. All other salamanders occurred in the culvert system within 22 m of a spring outlet. On average salamanders were found 2.5 m from a spring outlet, and most captures occurred within 1 m of an outlet. The spring run downstream of the box culvert possesses the shallow, flowing water and abundant rocks with interstitial gaps that is typical of central Texas Eurycea occupied springs (Sweet, 1982; Chippindale, 2005; Bowles, Sanders & Hansen, 2006), but it lacks any apparent groundwater gaining sections. In contrast, the cracks and openings in the culvert system provides spring discharge outlets but lacks the typical habitat structure, and often yard waste and litter are the only available cover objects. Other studies document that salamanders are more likely to utilize culverts if continuous rock substrate and cover objects are present because these features mimic natural streambeds (Ward, Anderson & Petty, 2008; Anderson et al., 2014), but at Brushy Creek Spring, proximity and access to the subterranean habitat appears more important. Downstream salamander distribution may be restricted by the pool between the gabion and spring run (Fig. 2D) which usually contains predatory fish thought to exclude E. tonkawae (Bendik, McEntire & Sissel, 2016).

It is highly unlikely that salamanders migrate to Brushy Creek Spring through surface water. The two closest known surface populations are Kreinke Spring (CHU 1) and PC Spring (CHU 7) which are approximately 7.5 km upstream in Brushy Creek and 14 km upstream in Lake Creek, respectively (U.S. Fish & Wildlife Service, 2013b). These portions of Brushy and Lake Creeks contain deep water, predatory fish, and large stretches without suitable E. tonkawae habitat. Further, we surveyed portions of Brushy Creek between Kreinke Spring and Brushy Creek Spring 38 times from 2013 to 2019 without observing salamanders, and we conducted over 50 surveys of portions of Lake Creek between PC Spring and Brushy Creek Spring from 2013 to 2019 without observing salamanders. Regular monthly to bi-monthly salamander monitoring has occurred at PC Spring from 2013 to 2021 and a salamander has never been observed more than 100 m from the spring outlets (ZC Adcock et al., 2021, unpublished data). Therefore, the most plausible source of E. tonkawae is subterranean animals immigrating into surface habitat. The entirety of our results indicates that rainfall causes increased spring discharge at Brushy Creek Spring which either expels resident subterranean salamanders (see Tovar & Solis, 2013) into the surface habitat or allows subsurface migration from another location.

Salamanders at this site predominantly occur within a structure that is property of, and maintained by, the City of Round Rock, although it occurs within Texas Department of Transportation Right of Way. The culvert system requires routine maintenance including debris removal and painting over graffiti. The findings of this study indicate that by carefully considering recent rainfall patterns in the area, maintenance could be scheduled to avoid instances when the presence of E. tonkawae within surface habitat is more likely. Similar findings have been highlighted for other amphibian groups, recommending road construction and maintenance be timed to avoid periods of high activity and movement (Hamer, Langton & Lesbarrères, 2015). Improved knowledge of the site-specific hydrogeology will help inform management of Eurycea occupied locations throughout the urban landscape in which they occur. One limitation to our study is the small sample size. Small sample sizes present difficulties in modeling/predicting the influence of parameters of interest. The question of how to properly manage a site occupied by threatened or endangered species for which only limited data exist deserves further investigation and discussion. Management of urban populations is necessary to prevent local extirpation of E. tonkawae, and improvement projects within this species’ CHUs are inevitable. We encourage additional studies aimed to inform anthropogenic activities to reduce incidental take of this listed species.

Conclusions

Brushy Creek Spring is one of the most anthropogenically modified sites occupied by E. tonkawae and is also a CHU for this federally threatened species (U.S. Fish & Wildlife Service, 2013b). Using exhaustive surveys of the most heavily modified portion of the site, we documented salamanders at Brushy Creek Spring for the first time in 21 years, negating the suggestion that salamanders had been extirpated from this locality (SWCA Environmental Consultants, 2012, unpublished report). The current physical structure of Brushy Creek Spring prevents accurate measures of groundwater discharge, but we overcame this issue by using local rainfall data to estimate recharge and to predict salamander relative abundance within surface habitat. Rainfall occurring 31–120 days prior to survey events best predicts salamander occurrence in the surface habitat, and on average 29 cm of rainfall within that time period is likely to lead to salamander detections. Potential incidental take can be reduced by scheduling maintenance activities during periods in which salamanders are unlikely to occur in the surface habitat.

Supplemental Information

Supplemental Information 1 Jollyville Plateau Salamander capture histories at Brushy Creek Spring.

Unique identifier (column A) and capture history over 39 survey events (columns B-AN) for 27 individual Jollyville Plateau Salamanders detected at Brushy Creek Spring.

Click here for additional data file.

Supplemental Information 2 Salamander occurrence data and associated rainfall accumulation, environmental predictors, and observer effort.

Salamander occurrence data, e.g., date, location, effort (columns A-G). Water chemistry data, e.g., temperature, pH (columns H-K). Rain accumulations (columns L-AF).

Click here for additional data file.

Supplemental Information 3 R script for statistical analysis.

R script for grooming data, analyzing data using generalized linear models, and generating predictions and plots.

Click here for additional data file.

We thank Alex Llewelyn, Anjana Parandhaman, Autumn Yaroz, Ben Friou, Bill Keitt, Christopher Franke, Craig Crawford, Jeremiah Leach, Josh Rusnak, Kelen Capener, Reed Leible, and Dr. Shashwat Sirsi for field assistance. We thank Cal Newnam and Andrew Blair for information on site history and ownership. We additionally thank Matthew Kitchen, George Brooks, and one anonymous reviewer, for their thoughtful comments that improved the manuscript.

Additional Information and Declarations

Competing Interests

Author Contributions

Animal Ethics

Field Study Permissions

Data Availability

The authors declare that they have no competing interests.

Zachary C. Adcock conceived and designed the experiments, performed the experiments, analyzed the data, prepared figures and/or tables, authored or reviewed drafts of the paper, and approved the final draft.

Andrew R. MacLaren conceived and designed the experiments, performed the experiments, analyzed the data, prepared figures and/or tables, authored or reviewed drafts of the paper, and approved the final draft.

Ryan M. Jones performed the experiments, authored or reviewed drafts of the paper, and approved the final draft.

Andrea Villamizar-Gomez performed the experiments, authored or reviewed drafts of the paper, and approved the final draft.

Ashley E. Wall performed the experiments, authored or reviewed drafts of the paper, and approved the final draft.

Kemble White IV conceived and designed the experiments, authored or reviewed drafts of the paper, and approved the final draft.

Michael R. J. Forstner conceived and designed the experiments, authored or reviewed drafts of the paper, and approved the final draft.

The following information was supplied relating to ethical approvals (i.e., approving body and any reference numbers):

Texas State University provided full approval of this research.

The following information was supplied relating to field study approvals (i.e., approving body and any reference numbers):

Texas Department of Transportation approved of this research occurring within their right of way.

The following information was supplied regarding data availability:

Raw salamander occurrence data and associated environmental predictors, and R code used to analyze these data are available in the Supplemental Files.

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
