# Peer review of "Predicting surface abundance of federally threatened Jollyville Plateau Salamanders (Eurycea tonkawae) to inform management activities at a highly modified urban spring"

_PeerJ, doi:10.7717/peerj.13359_

## Round 0.1 · original submission · Major Revisions

After receiving two thorough and thoughtful reviews, and considering the comments, suggestions, and issues they raise, I agree that this is a fascinating and valuable study and something that if given some further attention could find a home here at PeerJ.

Both reviewers bought up some excellent points about how the manuscript can be improved, and did so from two different, albeit equally critical perspectives. While Reviewer 2 has laid out a number of excellent ways to improve the write up, reframe the narrative a bit, and better contextualize your work - all facets that will no doubt help you refine and polish this manuscript to increase it impact and readability. Review 1, on the other hand, has some excellent questions and comments about the analysis, and raises some issue with how it was conducted and how it is presented and interpreted. For this reason, we will need you to pay close attention to the issues being raise regrading the analysis section.

Notably, since the reviewers were not able to examine your raw data or code, particularly given some if the issue with analysis they raised, we will need you to resubmit these with your next draft (should you decide to), so they can be reviewed as well.

In the meantime, and in prepared your revisions, please play close attention to all of the comments both reviewers have made (not just the line by line comments). The helpful suggestion from Reviewer 2 will advance your ability to bolster the narrative and convey your information more effectively. And the suggestions, comments, and issues raised by Reviewer 2 will improve your analysis, make it more defensible in the public eye, and allow for by evaluation of your study during the next review.

With the appropriate attention paid to furthering this study, both in how it is analyzed, explained, and presented, I feel this review process has a lot of potential to really make this manuscript shine!

I am looking forward to reading the next version.

Reviewer 1 ·

Excellent Review

This review has been rated excellent by staff (in the top 15% of reviews)
EDITOR COMMENT
This is a great example of a thorough and thoughtful review that provides some useful and critical guidance for improving a manuscript. Additionally it does so in a helpful way. This is the type of review the helps science move forward!

Basic reporting

My suggestions for improving basic reporting:
1. More details are needed for the CJS mark-recapture models. For example, did the authors use goodness-of-fit tests?
2. Is there evidence that using pigmentation for Eurycea is a reliable way to identify individuals? I’m surprised that the Discussion framed sampling the species as “rare” rather than “rarely on the surface”. There may be a large population below the surface and only rarely do individuals surface because of the conditions (e.g. being flushed from flow or because of changes in dissolved oxygen, conductivity, or temperature).
3. The code and data the authors supplied do not include the capture histories or analyses for the mark-recapture section so I could only evaluate the details in the main manuscript.

Experimental design

My suggestions for improving the experimental design:
1. The research question is defined in the Introduction, but I was confused why the authors chose to model how rain, dissolved oxygen, and conductivity affect salamander counts. Right after stating this in the Introduction, the authors describe why these three variables are likely related to each other. The authors should state clearly why they included variables if they predicted they were related (or being driven by the same underlying mechanisms). Further to the research question, please clearly state how research fills an identified knowledge gap.
2. More justification and explanation is needed for how the generalized linear models (GLMs) were fitted. Several variables were multicollinear or expected to be (e.g. dissolved oxygen and specific conductivity appear negatively correlated, and the Introduction establishes why some predictor variables are likely related).
3. Further, fixing missing values or "outliers" to a mean using imputation risks masking real patterns. The authors changed the minimum value of conductivity to the mean, but if the hypothesis is that conductivity affects surface activity of salamanders then this will mask a real pattern. With only 39 points and about half having a 0 value for the response, changing the values of predictors seems riskier than simply removing those observations from the model. I recommend the authors further explain why they imputed values. Testing whether those changes affect conclusions would further convince me this analytic choice is not leading to spurious results.
4. The model selection for the GLM considers various measures of rainfall accumulation (varying the days prior to the survey that are included). However, the authors did not compare models without some of the other variables. Because there are so few observations and that some variables are likely correlated, this might affect conclusions. For example, I tested a model that just included rainfall accumulation 1 to 90 days prior to the survey (the most supported rainfall variable), conductivity, and the offset predictor. That model had more support than the next best-supported model in the paper (AICc 4.53 units lower) and both conductivity and rainfall accumulation significantly and positively affected the number of salamanders detected. Why did the authors not consider model subsets with fewer variables given the low sample size (39 maybe less if missing values are removed) and high number of predictors (5)?

Validity of the findings

My main concerns over the validity of the findings:
1. Not all raw data are provided (mentioned above in Basic Reporting).
2. Some of the conclusions are not supported by the analyses presented by authors. For example, in the Conclusion section the authors say: “The findings of this study indicate that by carefully considering recent rainfall patterns in the area, maintenance could be scheduled to avoid instances when the presence of E. tonkawae within surface habitat is predicted.” The data and model predictions are in Figure 5. The model predicts 1 salamander will be detected when rainfall exceeds 32 cm in previous 90 days (where the red horizontal line crosses the model prediction). However, the raw data show that 12 of 17 surveys when salamanders were detected had less precipitation than this value. Thus, suggesting maintenance when precipitation values are less than 32 cm does not seem supported by the raw count data the authors present.

Additional comments

This short paper examined a small population of Eurycea living in an urban environment to 1) estimate detection and survival, and 2) test what environmental conditions predict the number of detected salamanders in a culvert. I think the manuscript is interesting and contains unique data. I applaud the authors on their efforts working with such a challenging system where animals are hard to detect. I think this paper can make a valuable contribution, but I have several major concerns (outlined in previous sections) about the analytic approaches and the conclusions drawn from them.

Below, I outline a few additional specific suggestions for improvement.

L43: I’d recommend clarifying that it is the United States of America’s Endangered Species Act and definition of “take”.

L123: Change “Teas” to “Texas”.

L176: The R package RMark sends a call to program MARK and I recommend the authors cite MARK in addition to the R package.

L182: What was the number of survey periods in your study? Did you use data from both the spring run and the box culvert? It seems by definition the capture probability would decrease after 2018 when the amount of habitat you surveyed decreased. I know you do not have much data, but why did you not test variables that would affect detection probability that you collected? For example, DO, SC, or a seasonal effect? You later state that Eurycea in Texas are known to change surface activity seasonally, so I expected that it would have been included in the CJS model.

L186: Did you conduct goodness-of-fit tests for the CJS model? Please report the results here.

Table 1. According to the Methods and Results, there were 39 surveys (17 with, 22 without salamanders). Why are these numbers different? Are these the number of surveys only considering those with the water measurements?

Table 3. Please update to limit significant digits (i.e. use number of digits that correspond to precision).

·

Excellent Review

This review has been rated excellent by staff (in the top 15% of reviews)
EDITOR COMMENT
This is a great example of a thorough and thoughtful review that provides some useful and critical guidance for improving a manuscript. Additionally it does so in a helpful and supportive way. Reviews, like this one, helps science move forward!

Basic reporting

Overall the manuscript is well written, very interesting, and as someone that works with endangered species I applaud the tremendous effort that must have been exerted to collect this data!

I think the main thing that can be improved is the discussion. Currently there is a major lack of flow. The first two paragraphs of the discussion read like they belong in the introduction. The following paragraph has a lot of repetition from the study-site description and not much of a link to your findings. The next paragraph is mainly focussed on the limitations of your study, but this usually comes later, after you have discussed all your findings. Then, when discussing rainfall models there is no mention of the unimportant predictors (and why they are unimportant). Lastly, the ‘conclusion’ section is actually ‘management implications’ and feels a bit disjointed from the rest of the text at present – I think it would be better to blend management implications into the previous paragraphs as appropriate, and have a more general conclusions paragraph that summarizes your findings and perhaps why your study is interesting to researchers who study other endangered amphibians.

My second main concern would be the lack of citations to relevant literature. There is a huge body of work that deals with the issue of low detection probabilities in wildlife populations, including amphibians. Here are some that you might consider to strengthen the paper:

Ficetola, G. F. (2015). Habitat conservation research for amphibians: Methodological improvements and thematic shifts. Biodiversity and Conservation, 24, 1293–1310.
Gu W.D. and Swihart R.K. (2004) Absent or undetected? Effects of non-detection of species occurrence on wildlife habitat models. Biol Conserv 116:195–203.
Kellner, K.F. and Swihart, R.K. (2014). Accounting for imperfect detection in ecology: a quantitative review. PloS one, 9(10), p.e111436.
MacKenzie, D.I., Nichols, J.D., Sutton, N., Kawanishi, K. and Bailey, L.L. 2005. Improving inferences in population studies of rare species that are detected imperfectly. Ecology, 86(5), pp.1101-1113.
Mazerolle, M.J., Bailey, L.L., Kendall, W.L., Royle, J.A., Converse, S.J. and Nichols, J.D. 2007. Making great leaps forward: accounting for detectability in herpetological field studies. Journal of Herpetology, 41(4), pp.672-689.


Similarly, there is a wealth of studies that investigate the environmental correlates of amphibian surface activity that would be good to reference in the intro and/or discussion. For example:

Canavero, A., M. Arim, D. E. Naya, A. Camargo, I. D. Rosa, and R. Maneyro. 2008. Calling activity patterns in an anuran assemblage: the role of seasonal trends and weather determinants. North-West Journal of Zoology 4: 29–41.
Jaeger, R.G., 1980. Microhabitats of a terrestrial forest salamander. Copeia, pp.265-268.
Sanchez, K., Grayson, K.L., Sutherland, C., Thompson, L.M. and Hernandez-Pacheco, R., 2020. Environmental drivers of surface activity in a population of the Eastern red-backed salamander (Plethodon cinereus). Herpetological Conservation and Biology, 15(3), pp.642-651.
Semlitsch, R. D. 1985. Analysis of climatic factors influencing migrations of the salamander Ambystoma talpoideum. Copeia 1985: 477–489.
Sexton, O. J., C. Phillips, and J. E. Bramble. 1990. The effects of temperature and precipitation on the breeding migration of the spotted salamander (Ambystoma maculatum) Copeia 1990: 781–787.
Taub, F.B., 1961. The distribution of the red-backed salamander, Plethodon c. cinereus, within the soil. Ecology, pp.681-698.

Experimental design

The research question is well defined and it is clearly stated how this research fills a knowledge gap. The methods are described with sufficient detail to replicate. My only suggestion would be to explore simpler models for the negative binomial count models, i.e., with just rainfall as a predictor. I feel there is justification to remove all other environmental covariates post-hoc.
Also there are some t-tests performed in the accompanying R script that are not described, and those results not included.

Validity of the findings

All underlying data have been provided, and an associated R script. However the code for the mark-recapture analysis is not included in the script. The R script also contains some t-tests which would be great to describe in the methods section and add to Table 1.

As I stated in ‘Basic Reporting’, the discussion could use some revision to improve clarity and flow. If this is addressed it will be much easier for readers to grasp the main findings and importance of the study.

Additional comments

I would include one or two sentences in the abstract stating the main findings (even if they are general)

Line 20-21: “are a range restricted” should be “is a range restricted”

Line 61-62: I would suggest moving this last sentence to the end of the next paragraph (~line 77), as it seems to fit better there, once you’ve explained the extent of urban structures in the CHUs.

Line 89-90: I would suggest moving this last sentence to come before the previous two (~line 86; i.e., so that if follows directly on from the sentence “Additionally, the site was drastically altered…”). I think this would improve the flow of the paragraph.

Line 92-93: I would name the site again for clarity. And I don’t think it’s necessary to state that your “goals evolved”. So the sentence would just be: “Here we report the findings of seven years of surveys at Brushy Creek Spring.“

Line 93-94: this first aim seems odd when in the previous paragraph you have stated that these salamanders can be “regularly detected” at this site. I think you either need to remove this aim, or revise the previous paragraph to make it clearer that the site status was unknown before this study (which is a cooler angle!).

Line 103-114: I think this overview of environmental conditions on salamander abundance/detectability should be its own paragraph, and come sooner in the introduction. You should try to limit the last paragraph to the aims/hypotheses/predictions of this study.

Line 123: “Texas” is misspelled.

Line 151-154: I would move the permit information to the end of this section (i.e., after you have described capture and measuring protocols)

Line 161: I’m not sure you ever used gravidity status in any of your analyses, so I don’t think it’s worth its own figure. I would rather see more results-based figures - at present there is only one.

Line 165-173: this environmental data could be a separate paragraph, or might be better attached to the end of the first paragraph in this section.

Line 244-245: this sentence belongs in the results

Line 248: when you found that day of year, DO, and conductivity did not influence salamander counts, why did you not drop them from the models? What is the AIC of simpler models (i.e. with only rainfall as a predictor)?

Line 286-287: this sentence belongs in the discussion

Line 289-290: This is incorrect – it is the difference in AIC values that is less than 2 (delta-AIC), not the AIC values themselves.

Line 307-329: I feel like these first two paragraphs belong in the introduction. This is background info that would help justify the present study. If you would like to keep it in the discussion, it definitely shouldn’t be the opening paragraphs. The first paragraph is typically a summary of your main findings and a statement about the relevance/importance of the study (i.e., how it will inform management/conservation).

Line 332: It seems like a missed opportunity to me that you emphasize the importance of “distance from spring outlet” for detecting salamanders, but you didn’t include anything related to this in your models… I feel this would be more informative than including the various rainfall models that all say roughly the same thing. Not sure if this is possible, but a figure showing the declining probability of detection with distance from spring outlets would be really cool!

Line 334: inconsistency between “spring-run” and “springrun”. Not sure which one is correct, but check document and stick to one of them.

Line 345-346: There is a huge body of literature on dealing with low detection probabilities in wildlife studies. This paragraph is fine to keep but should include lots of references.

Line 350-351: I know this is beyond the present study, but this type of uncertainty can be dealt with – either in a Bayesian framework where you place an informative prior on detection probability, or using a mark-recapture model that allows for temporary emigration. Below are some relevant citations:
Bailey, L.L., Simons, T.R. and Pollock, K.H., 2004. Estimating detection probability parameters for plethodon salamanders using the robust capture‐recapture design. The Journal of Wildlife Management, 68(1), pp.1-13.
Kendall, W.L. and Bjorkland, R., 2001. Using open robust design models to estimate temporary emigration from capture—recapture data. Biometrics, 57(4), pp.1113-1122.

Line 390: there is no mention in the discussion concerning the lack of importance of ‘day of year’, DO, and conductivity, but to me this is an important result, especially in the context of management implications. I would add a paragraph speculating on why these predictors were not important.

Table 1 title: “are” should be “at”

Table 1: The accompanying R script has various t-tests for these comparisons, so I think this table could be modified to include the results of these tests, e.g., separate columns for ‘Temp Present’ and ‘Temp Absent’ and a column for t-test result. Also the t-tests should be described in the methods.

Table 3 legend: just as an FYI, an equation is not displaying properly on the pdf

Figure 3: I’m not sure this figure is needed/relevant

Figure 5: I feel like you have a lot more non-detections in your dataset… are they not plotted or is there a lot of overlapping points? If there is an easy way to make this clearer that might be nice. Overlapping can maybe be fixed with geom_count() or jitter().

---

## Round 0.2 · Minor Revisions

The author did an excellent job revising the manuscript and with a few small tweaks, I think I would be happy to recommend that it be accepted.

Please see the comment for a minor edit from a reviewer and the attached pdf where I was able to catch a few typos and spot additional punctuation (which are mostly suggestions). Our system requires I upload a pdf, so just have a look through and see where I have removed double spacing, added spaces, or commas - that sort of thing - they should be marked in red.

Once you are happy with it, after looking over the suggested changes, just send back a cleaned version and we can take it from there.

This is a neat study on a curious salamander, and I think people will enjoyed learning more about them (as I did).

Reviewer 1 ·

Basic reporting

The revised manuscript has an improved article structure, includes all relevant raw data, and the figures are improved.

Experimental design

No comment.

Validity of the findings

All underlying data is now included. The conclusions are well stated in relation to original research question.

Additional comments

The authors did a great job revising the manuscript and have addressed all my previous major concerns and minor suggestions.

A few very minor suggestions:

L319: You can remove "(∆AICc)" because it is redundant.

L333: Change "The competing models mean" to "The competing model's mean".

L369-372: This is a valuable discussion on the lack of recaptures. I recommend at least mentioning that "survival" is estimated as "apparent survival", which cannot disentangle deaths from permanent emmigration. For example, individuals may be either swept downstream or permanently retreat to subsurface habitat.

---

## Round 0.3 · accepted · Accept

I am please to recommend that this manuscript be accepted. You all did a great job addressing some of the helpful comments and suggestions from the reviewers, and I think this manuscript is looking quite nice. Well done on all your hard work! Both in conducting the study and on drafting this article. I am looking forward to seeing this in 'print' soon and hearing more about this curious species in the coming years. Congratulations!